# The Impact of Different Withering Approaches on the Metabolism of Flavor Compounds in Oolong Tea Leaves

**DOI:** 10.3390/foods11223601

**Published:** 2022-11-11

**Authors:** Yahui Wang, Chenxue Li, Jiaqi Lin, Yun Sun, Shu Wei, Liangyu Wu

**Affiliations:** 1College of Horticulture, Fujian Agriculture and Forestry University, Fuzhou 350002, China; 2State Key Laboratory of Tea Plant Biology and Utilization, Anhui Agricultural University, Hefei 230036, China

**Keywords:** oolong tea, withering, metabolomics, proteomics, flavor compounds

## Abstract

In this study, complementary metabolomic and proteomic analyses were conducted on the solar- and indoor-withered oolong tea leaves, and freshly plucked leaves as the control, for the purpose to reveal the mechanisms underlying the initial formation of some flavor determinants during the early stage of oolong tea processing. As a result, a total of 978 non-volatile compounds and 152 volatile compounds were identified, the flavonoids and several esters were differently accumulated in various tea samples. In total, 7048 proteins were qualitatively and quantitatively determined, the analysis on pathway enrichment showed that phenylpropanoid, flavonoid metabolisms, and protein processing in endoplasmic reticulum were the major pathways discriminating the different tea samples. The joint protein–metabolite analysis showed that the multiple stresses such as dehydration, heat, and ultra-violet irradiation occurred during the withering step induced the dynamic and distinct changes in the biochemical network in the treated leaves compared to fresh leaves. The significant decreases in flavonoids, xanthine alkaloids, and several amino acids contributed to the alleviation of bitter or astringent taste of withered leaves, although the decomposition of L-theanine resulted in the loss of umami flavor over the solar-withering step. Moreover, the fruity or floral aromas, especially volatile terpenoids and phenylpropanoids/benzenoids, were retained or accumulated in the solar withered leaves, potentially aiding the formation of a better characteristic flavor of oolong tea made by indoor withered tea leaves. Distinct effects of solar- and indoor-withering methods on the flavor determinant formation provide a novel insight into the relationship between the metabolite accumulation and flavor formation during the withering step of oolong tea production.

## 1. Introduction

Oolong tea, one of the most consumed teas worldwide, is famous for its brisk taste and floral aroma. The annual production of oolong tea in China was 287,200 tons in 2021, accounting for 8.5% domestic sales [1], and the popularity of oolong tea is primarily attributed to its unique characteristics formed during the manufacture process [2]. Typically, the production process of oolong tea can be summarized as the following steps: plucking, withering, fermentation, panning, rolling, and drying. First, the banjhi tea shoots composed of two or three leaves are collected and subjected to the sequential manufacturing process, of which the withering is the initial step of the postharvest production of oolong tea.

As the first step in oolong tea processing, withering is crucial to initiate the formation of some oolong tea flavor determinants during the production of oolong tea. Moreover, withering leads to substantial water loss of the freshly plucked leaves and turns them soft so that the subsequent processing steps can be carried out for the further development of the distinctive taste and aroma of oolong tea [3]. For withering, freshly plucked tea shoots are spread on mats under the sunlight for 30–45 min, then moved indoors, and setting down for 30–60 min according to the canonical manufacturing process [4]. Natural sunlight withering is an indispensable and inexpensive way to quickly turn tea shoots dehydrated and wilted for further processing [5]. Nevertheless, it is known now that during the withering step, the leaves are subjected to multiple stresses including dehydration, heat, or ultra-violet (UV) irradiation, resulting in the slightly shrunk or wilted appearance. These stresses also reprogram numerous intracellular enzyme-catalyzed biological processes such as photosynthesis, transpiration, and metabolism, consequently resulting in extensive metabolomic changes in the leaf tissues [6]. Sunlight serves not only as an environmental stress factor, but also as a biological stimulant to regulate the accumulation of flavor metabolites, in addition to providing the energy for various biological activities [7]. For instance, sunlight can effectively induce the accumulation of aromas, especially those in the terpenoid metabolic pathway [8]. Moreover, under solar withering, several microRNA modules activated in the tea leaves suppress flavonoid biosynthesis, but boost the terpenoid biosynthesis [9]. The levels of phytohormones such as abscisic acid, jasmonic acid, and salicylic acid have been found to be dramatically altered in response to the dehydration stress within 24 h indoor withering, potentially impacting the contents of gallocatechin and epicatechin [10]. Moreover, during white tea indoor withering, the total contents of catechins and starch continuously decrease, by contrast to the contents of theaflavin, γ-aminobutyric acid (GABA), maltose, and soluble sugars significantly increase [11]. Although the natural solar withering could promote the pleasant fragrance of oolong tea, the indoor withering is a universal alternative in withering step in practice of oolong tea production [12], a recent study showed that the contents of total flavonoid, catechins, and several volatiles were highly retained in indoor-withered tea leaves, regulating the quality of oolong tea in a different way [13]. 

To date, it has been found that multiple stresses during solar or indoor withering promote the special flavors by inducing significant changes in the volatile and non-volatile metabolites under the catalysis of enzymes [8,14,15]. However, limited attention has been paid to elucidate the relationship between the metabolic changes induced by different withering approaches and subsequent flavor formation in the tea leaves during oolong tea manufacturing. 

Recently, the multi-omics technique has been increasingly used to reveal the complex biological processes in organisms. The metabolomic analyses using mass spectrometry are employed to quantify the metabolites, while the proteomic determination is applied to detect the protein expression in plants. A combination of metabolomics and proteomics techniques could be a promising approach to elaborate the dynamic changes in key flavor metabolite formation and enzyme–metabolite interplay over the production process of various teas [16,17]. Thus, in this study freshly plucked tea shoots were grouped into two sets for solar- and indoor-withering treatments, respectively. The aim of this work was to illustrate corresponding changes in the metabolomic and proteomic profiles in order to better understand the enzymatic regulation of the dynamic changes in volatile and non-volatile compounds over the different withering steps, the results will shed light on dynamic changes in metabolites in response to multiple environmental stresses during the withering step and contribute to a better understanding of the mechanisms in the flavor formation of oolong tea.

## 2. Materials and Methods

### 2.1. Tea Leaf Processing 

The banjhi tea shoots consisted of two or three leaves that were harvested from *Camellia sinensis* cultivar Fujian Shuixian grown on Juyuan tea plantation in Zhangping City (25°28′50″ N, 117°20′16″ E, altitude: ~520 m) in October 2020. The tea plants used in this study were cultivated conventionally with regular fertigation. After plucking, the fresh leaves were withered for 30 min respectively under the sunlight (SW) and indoor (IW) conditions (Figure 1). Photosynthetic active radiation for SW and IW were 550 ± 30 μmol/(m^2^·s) and 0 μmol/(m^2^·s), respectively. Then, the tea shoots either in SW or in IW were moved into the same workshop and set down for 15 min; afterwards, the tea shoots from SW and IW groups, together with the fresh leaves just after plucking (control, CK), were used for sampling. For sampling, the second leaf basipetal from the bud in one given shoot (randomly selected) was cut off with scissors and immediately immersed in liquid nitrogen for 5 min, then placed in dry ice for transportation prior to storage at −80 °C in a refrigerator (DW86L626, Haier, Qingdao, China). Each sample was composed of three biological replications. 

### 2.2. Widely-Targeted Metabolomic Analysis on Non-Volatile Metabolites in Tea Samples 

The non-volatile metabolite extraction and analysis were performed by a commercial service company (MetWare Company, Wuhan, China) according to the method reported previously [14] with some minor modifications. Briefly, the samples were freeze-dried and ground into powder using a mill (MM 400, Retsch Company, Haan, Germany) for 120 s at 35 Hz. Then, 100 mg powder of each sample was weighed and extracted using 1 mL 70% methanol containing 0.1 mg L^−1^ lidocaine as the internal standard at 4 °C overnight, and centrifuged at 10,000× *g* for 20 min at 4 °C. The supernatant was collected and filtrated by 0.22 μm filter (Millipore Corp., Bedford, UK) prior to further detection. 

The aqueous solution analyses were carried out on an ultra-performance liquid chromatography electrospray ionization tandem mass spectrum (UPLC-ESI-MS/MS) system (UPLC: Shim-pack UFLC SHIMADZUCBM20A; MS/MS: Applied Biosystems 4500 QTRAP; Thermo Fisher Inc, Waltham, MA, USA). Identification and annotation of metabolites were performed by comparing the retention time, mass-to-charge (m/z) values, and the fragmentation patterns with authentic standards or by searching against the internal database and public databases (KNApSAcK, MassBank, MoTo DB and METLIN). 

### 2.3. Volatile Compound Analysis in Tea Samples

Qualitative and quantitative analysis on volatile metabolites was according to our previous report [18]. In short, the tea samples were ground into powder in liquid nitrogen, and then an aliquot of the powder (1.0 g) was immediately transferred to a 20 mL head-space vial (Agilent Technologies, Palo Alto, CA, USA) containing 400 μL of 20% NaCl saturated solution to inhibit any enzyme reaction. The 2,6-Dimethyl-4-heptanone was added as an internal standard to a final concentration of 0.125 ng μL^−1^. The vials were sealed using crimp-top caps with Agilent TFE/silicone headspace septa (Palo Alto, CA, USA), then subjected to solid-phase micro-extraction (SPME). For SPME analysis, each vial was incubated for 10 min at 60 °C, followed by the exposure to a 65 µm divinylbenzene/carboxen/polydimethylsilioxan fiber (Supelco Analytical Corporation, Darmstadt, Germany) in the headspace of the sample for 20 min at 60 °C. Then, the desorption of the volatile compounds from the fiber was performed in an injection port of 7890B gas chromatography apparatus (Agilent Technologies, Palo Alto, CA, USA) for analysis. The volatile metabolites were identified by comparing the mass spectra with the data system library (MWGC or NIST) and the linear retention index. 

### 2.4. Identification and Quantification of Proteins

The protein extraction, digestion and data-independent acquisition (DIA) analysis were performed by a commercial service company (Gene de novo Biotechnology, Guangzhou, China) according to our previous study with a minor modification [19]. In brief, the tea samples were ground in liquid and then mixed with 2.5 mL of lysis-buffer composed of 8 M urea, 2% sodium dodecyl sulfate, and 1× Protease Inhibitor Cocktail (Roche Ltd. Basel, Switzerland), subsequently, the mixture was ultrasonicated on ice for 40 min and centrifugated at 10,000× *g* for 40 min at 4 °C, the supernatant was collected and precipitated in ice-cold acetone for 12 h. Then, the precipitation was washed with pre-chilled acetone three times and dissolved in 8 M urea through sonication on ice. The quality of protein was assessed by sodium dodecyl sulfate polyacrylamide gel electrophoresis (SDS-PAGE), and the concentration of protein was determined by bicinchoninic acid disodium protein assay kit (ThermoFisher Scientific Inc., Waltham, MA, USA). The obtained protein (50 μg) was re-suspended in 50 μL of 8 M urea, followed by adding 1 μL of 1 M dithiotreitol, and then the mixture was incubated at 55 °C for 65 min. Subsequently, the mixture was further mixed with 5 μL iodoacetamide (20 mM) at 37 °C for 60 min in the dark. Afterwards, the sample was precipitated in 300 μL of prechilled acetone at −20 °C for 12 h. The precipitate was washed twice with cold acetone, and then re-suspended in 50 mM ammonium bicarbonate. The obtained proteins were digested at 37 °C for 16 h with modified trypsin (Promega Co, Ltd., Madison, WI, USA.) with the substrate/enzyme ratio of 50:1 (*w*/*w*). The peptide mixture was dissolved in buffer A (20 mM ammonium, formate in water, Ph = 10.0), and then fractionated by Ultimate 3000 system using a reverse phase column (XBridge C18 column, 4.6 mm × 250 mm, 5 μm, Thermo Fisher scientific, Waltham, MA, USA). The gradient elution condition was as follows: starting from 95% buffer A/5% buffer B to 55% buffer A/45% buffer B (20 mM ammonium formate in 80% acetonitrile, pH = 10.0) within 42 min, flow rate 1.1 mL min^−1^, column temperature 28 °C. The eluted peptides were separated into 10 fractions and each fraction was dried in a vacuum concentrator for the next step. 

The peptides were re-suspended with 30 μL of buffer C (0.1% formic acid in water) and determined by online LC-MS/MS on an Orbitrap Fusion Lumos coupled to EASY-nLC 1200 system (Thermo Fisher Scientific, Waltham, MA, USA). The gradient elution condition was as follows: from 95% buffer C/5% buffer D (0.1% formic acid in acetonitrile) to 65% buffer C/35% buffer D within 120 min, flow rate 300 nL min^−1^. The raw data of DIA analysis were processed and quantified by Spectronaut Pulsar 11.0 (Biognosys AG, Zurich, Switzerland) with default parameters, the normalized expression level of a given protein was represented as the median of all unique peptides annotated as the same protein. The different regulated proteins (DRPs) were filtered with the selection criteria of fold change > 1.5 or <0.67 and false discovery rate (FDR) < 0.05. The identified proteins were annotated by gene ontology (GO), and Kyoto Encyclopedia of Genes and Genomes (KEGG) databases to acquire their functions. DRPs across various groups were screened by using R package (http://www.rproject.org/, accessed on 20 January 2020). 

### 2.5. The Electronic Tongue (E-Tongue) Analysis on the Tea Samples

The taste attributes of the tea samples were assessed on an E-tongue apparatus (TS-5000Z, Insent Electricity Company, Tokyo, Japan). Briefly, an aliquot of a tea sample (3.0 g) was brewed with 150 mL freshly boiled water for 300 s, and subsequently the tea brewing was filtered with 0.45 μm filter (Therom Fisher, Waltham, MA, USA) and cooled to room temperature for the following assessment. The E-tongue apparatus is composed of several sensor probes and a reference probe, detecting the taste intensity by dipping the sensor probes into the tea brewing and reference solution. Prior to the beginning of the experiments, a sensor check was conducted routinely before every measurement in order to assure that sensors were stable in the correct mV range. One measurement cycle consisted of measuring a reference solution (Vr), followed by the sample solution (Vs), a short (2 × 3 s) cleaning procedure, and measurement of the aftertaste (Vr’) in reference solution. The aftertaste was measured by determining the change in membrane potential caused by the substance adsorption to the lipid membrane after the short cleaning procedure. Both sensor output for taste, also called relative value (R), and sensor output for aftertaste, also called CPA value (change of membrane potential caused by adsorption), were calculated in relation to the preliminary determined sensor response to the reference solution (Vr). The formula used was R = Vs − Vr; CPA = Vr’ − Vr. The acquired taste dataset was used for further analysis to describe the taste attributes of the tea samples.

### 2.6. Data Analysis

The tea samples composed of three replicates were used for the E-tongue, metabolomics, and proteomic determination. The peak area of each metabolite was normalized to the peak area of internal standard prior to further data processing. The principal component analysis (PCA) was carried out on Origin2018 (OriginLab Corporation, Northampton, MA, USA). The hierarchical cluster analysis (HCA), as well as Orthogonal projection to latent structures discriminant analysis (OPLS-DA) were run on Simca (Umetrics Company, Malmö, Sweden) for the statistical analyses. The variable importance projection (VIP) values were calculated to distinguish the major differential metabolites with criteria of VIP values ≥ 1.0 and log_2_|(fold change)| ≥ 1 (*p* < 0.05) for each pair comparison.

## 3. Results and Discussion

### 3.1. Changes in the Non-Volatile Metabolites during Withering Process

To investigate the changes in metabolites in the tea samples over the different processes, widely targeted metabolic analysis was performed to profile methanol-soluble extraction from SW, IW, and CK samples. As a result, a total of 978 non-volatile compounds were identified, and the metabolomic data were normalized and visualized to reveal the difference in various samples as shown in Figure 2A, in which the constitutes in the withered tea samples (SW and IW) were dramatically distinct from those in the fresh leaves (CK), indicating that the plucked tea leaves suffered from heat, dehydration, or solar irradiation, resulting in the changes in the intracellularly biological processes and metabolites accumulation. Clearly, the triplicates of the same tea sample were gathered and different samples were separated from each other (Figure 2B). The first two principal components (PCs) accounted for 78.1% of the total variance (PC1 = 56.4%, and PC2 = 21.7%). Obviously, the samples from the CK group distributed at negative direction of PC1, in contrast to those of SW and IW; the IW samples were located in the first quadrant, while the SW samples were located in the fourth quadrant, which were distributed at the opposite directions of PC2 (Figure 2B). The distinct distributions of tea samples on the PCA plot are in line with the observation of the different metabolite abundance of the various samples in Figure 2A. 

Subsequently, we refined the differentially regulated non-volatiles (DRNs) with the criteria that VIP values ≥ 1.0 and log_2_|(fold change)| ≥ 1 (*p* < 0.05) in paired comparisons. As presented in Figure 2C,D, the most significantly changed amounts of DRNs were observed in SW/CK (61 up-regulated, 32 down-regulated), followed by IW/CK (38 up-regulated, 54 down-regulated), and SW/IW (28 up-regulated, 6 down-regulated). This result suggests that the alterations of non-volatile components in post-harvested tea leaves suffered from multi-stress were induced to various conversions within withering step, which is a systematic response to adverse environment conditions [20].

To better understand the dynamic changes in metabolites during the withering process, the DRNs that shared in both SW/CK and IW/CK groups were identified, then subjected to the pathway clustering analysis by using online platform MetaboAnalyst (https://www.metaboanalyst.ca/, accessed on 15 May 2022). A total of 56 DRNs were obtained as presented in Figure 2E and Appendix A. Roughly, these DRNs could be divided into two groups according to the dynamic changes, with 27 DRNs higher in CK group, 29 DRNs lower in CK group (Figure 2E). The top three enriched pathways were flavonoid biosynthesis, followed by flavone and flavonol biosynthesis, and glutathione metabolism (Figure 2F), suggesting that the multiple stresses, including sunlight exposure, heat, or dehydration over the SW or IW process potentially induced the superfluous accumulation of reactive oxygen species (ROS), spontaneously triggering the biosynthesis of antioxidants such as glutathione or flavonoids in response to the ROS stress [21,22]. The pathways “Flavone and flavonol biosynthesis” were enriched over the withering step (Figure 2F), closely associated with the accumulation of the non-volatile compounds [14]. The contents of flavones and flavonols, such as luteolin, ayanin, apigenin, and myricetin, in withered leaves were decreased comparing to those in the CK group (Figure 2F), which is probably depleted as antioxidants in response to the oxidative stress caused by the adverse environment, contributing to the lowering of the bitter palate in oolong tea [18]. The glutathione, derived from glutamic acid and cysteine under the catalysis of a series of enzymes, is involved in several physiologic processes in plants under stress conditions. The adverse condition including heat, UV irradiation, and water loss, occurring over the sunlight withering and indoor withering steps would trigger the cellular oxidative stress, inducing the consumption of stored glutamic acid, lowering the umami taste of tea as reported in the previous study [18]. The DRNs in the SW/IW group were also obtained and then subjected to the pathway analysis, to explore the difference in non-volatile accumulation between SW and IW groups. A total of 34 DRNs were differentially accumulated in SW and IW groups (Figure 2G, Appendix A), with 28 DRNs higher in SW sample, indicating that tea leaves responded differentially to sunlight and indoor withering treatments due to their different stress conditions, as expected. Phenylalanine is generated from the chorismic acid pathway, then channels into the phenylpropanoid biosynthesis. As the derivatives of phenylalanine and phenylpropanoid biosynthesis pathways, the contents of *p*-coumaryl alcohol, 2-hydroxycinnamic acid, α-hydroxycinnamic acid, and 3-(3-hydroxyphenyl)-propionic acid were higher in SW than in SW (Figure 2G). The increasing levels of these compounds was probably associated with the heat and dehydration stresses under the sunlight, since these derivatives are involved in the plant resistance to UV radiation, extreme temperature, and continuous drought threats [23]. Moreover, α-linolenic acid metabolism was also significantly enriched, owing to the different accumulation of jasmonic acid and methyl jasmonate in SW and IW (Figure 2G). This observation was in line with our previous result showing that the accumulation of jasmonic acid derivatives played a signal to activate downstream genes in response to dual stresses [18]. The result of pathway clustering analysis showed that glutathione metabolism was also enriched (Figure 2H), implying that the multi-stress occurring during the solar withering aroused the stress response in tea leaves, giving rise to the accumulation of antioxidants. 

### 3.2. Changes in the Volatile Metabolites during Withering Process

In total, 152 volatile compounds were identified and determined from the SW, IW and CK samples (Figure 3A), which also showed marked difference among the three samples (Figure 2A). The result of PCA analysis showed that the three sample groups could obviously be discriminated from each other based on their volatile profiles, and the triplicates of each tea sample were gathered (Figure 3B); moreover, the first two principal components (PCs) accounted for 64.6% of the total variance (PC1 = 39.3%, and PC2 = 25.3%), indicating that the volatile metabolomic dataset of each group showed dynamic changes according to the different treatments, and the repeatability of the result was reliable. To obtain the differentially regulated volatiles (DRVs) among the different tea samples, the volatile components were compared and the DRVs were screened out with the criteria of VIP values ≥ 1.0 and log_2_|(fold change)| ≥ 1 (*p* < 0.05) in paired comparisons. As shown in Figure 3 C and D, the least number of DRVs were observed in IW/CK, with the most in SW/CK. Notably, all of the DRVs in SW/IW comparison were up-regulated (Figure 3C), indicating that sunlight-withering promoted the accumulation of volatile compounds in oolong tea leaves, which is consistent with the observation that solar irradiation facilitated the characteristic aroma of oolong tea [9]. 

A total of 11 shared DRVs, composed of 1 ketone, 3 esters, 1 olefin, 4 terpenes, 1 aromatic, and 1 heterocyclic compound, were identified in both SW/CK and IW/CK groups (Figure 3E, Appendix A). It is worth noting that three esters, namely, hexyl acetate, hexyl butyrate, and hexyl hexanoate, were higher in SW and IW than in CK. The previous studies showed that the polyunsaturated fatty acids are converted into C_6_ and C_9_ aldehydes under the catalysis of hydroxyperoxide lyase, which would be reduced to alcohols by alcohol dehydrogenases in response to environmental stress, then further converted to their esters [24,25]. In this study, the hexyl esters in tea leaves were potentially massively accumulated by the induction of solar withering after harvesting, leading to the enrichment of fruity fragrance in sun-withered tea leaves, since these esters are the crucial contributors of apple-like fruity aroma [26,27]. The DRVs in the SW/IW are presented in Figure 3F; there were 29 volatiles (18 esters, 3 alcohols, 2 ketones, 2 terpenes, 1 acid, 1 aldehyde, 1 aromatics and 1 heterocyclic compound, Appendix A), identified as the biomarkers between the two samples with different withering methods. As the most abundant volatiles in solar-withered leaves, the esters were the major source of floral or fruity fragrances [28], suggesting that sunlight withering could significantly elevate the abundance of characteristic fragrance in tea leaves compared to the indoor-withered leaves, which agrees with the observation that short-term sunlight exposure promotes the accumulation of aromatic substances in the postharvest leaves of oolong tea in contrast to indoor withering [8]. Apart from the esters, the α/β-ionone and jasmine lactone also dramatically contributed to the floral odor of oolong tea (Figure 3F), especially the β-ionone, which would be massively produced during oolong tea solar withering. This observation suggests that the solar exposure caused the damage in leaf marginal cell, leading to the enhancing activity of carotenoid cleavage dioxygenase and fast accumulation of β-ionone under sunlight exposure [29,30,31], indicating that the sunlight withering facilitated the accumulation of the characteristic aroma of oolong tea in this study. 

### 3.3. Changes in the Proteins during Withering Process

A total of 7048 proteins were identified and quantified via DIA determination (Figure 4A). The PCA analysis on the proteomic profile showed that the SW, CK, and IW groups were located in first, third, and fourth quadrant, respectively; additionally, the different samples were distinguished from each other, and the three biological replicates were relatively clustered, with the first two PCs accounting for 56.0% of total variance (Figure 4B), demonstrating a fine representativeness of our samples. The DRPs in the given pair comparison were predicted with the criteria of fold change > 1.5 or <0.67 and FDR < 0.05. As shown in Figure 4C,D, the SW/CK possessed the most amount of DRPs (169 up-regulated, 234 down-regulated), followed by IW/CK (93 up-regulated, 143 down-regulated), and SW/IW (35 up-regulated, 80 down-regulated). In general, postharvest treatments on tea leaves involving UV irradiation, heat, and dehydration, trigger adaptation mechanisms such as protein degradation, which plays an integral role in cell physiology and development by eliminating abnormal proteins and short-lived regulators [32]; additionally, the adverse conditions could also result in the biosynthesis of proteins that associated with cellular maintenance and osmoprotection against oxidative stress [33]. In this study, the varied changes in pair comparison indicated that the multi-stress including UV irradiation, heat, and dehydration occurred in solar-withered leaves triggered a series of sophisticated alterations on protein level, which is similar to the previous study that sunlight withering promotes the alternative splicing on mRNA to those of indoor withering or fresh leaves [34]. Together with the analyses of different regulated metabolites, the conversions of flavoring compounds over the oolong tea withering were temporarily intensified under the catalysis of enzymes when the leaves are suffering from multiple stresses, resulting in the stepwise formation of the characteristic taste or fragrance of oolong tea [35]. 

To explore the impact of withering on the changes in proteins, the shared DRPs in both SW/CK and IW/CK comparisons were filtrated for further investigation. As shown in Figure 4E, a total of 125 proteins were significantly altered in both pair comparisons (Appendix A). Subsequently, the analysis on the pathway enrichment of the DRPs was performed according to the KEGG biological network. As a result, the DRPs in SW/CK was enriched in the pathways of phenylpropanoid biosynthesis, flavonoid biosynthesis, phenylalanine metabolism, DNA replication, nitrogen metabolism, protein export, starch and sucrose metabolism, phagosome, base excision repair, photosynthesis according to the rich factor (number of matched proteins/number of background proteins), and −Log_10_*P* (Figure 4F). As the precursors of flavonoids, phenylpropanoids are a cluster of plant secondary metabolites originating from phenylalanine, functioning as structural or signaling molecules; moreover, the cascades of phenylpropanoid branches among various downstream mechanisms depend on the metabolic flux redirection, demonstrating a combination of complexity and flexibility in response to environmental stresses [36]. In this study, the solar-withered tea leaves subjected to the UV irradiation, dehydration, and heat spontaneously stimulated the resistance system, such as phenylpropanoid and flavonoid biosynthesis, to cope with the adverse environmental stimuli. In addition to flavonoid biosynthesis, phenylalanine metabolism, and phenylpropanoid biosynthesis, the DRPs in IW/CK were dramatically enriched in the spliceosome (Figure 4G). This observation is in line with the previous reports, in which the different expressed genes was clustered in the spliceosome under mild dehydration stress, and the spliceosome complex or splicing-related proteins were dominant DRPs in response to drought [37,38]. There were 115 DRPs obtained from SW/IW comparison (Figure 4H), and the pathway analysis showed that the protein processing in endoplasmic reticulum was the most enriched pathway, followed by phenylpropanoid biosynthesis and galactose metabolism (Figure 4I). As presented in Appendix A, the substantial number of DRPs belongs to heat shock proteins, indicating stress-activated pathways were involved over the solar withering, which is agreed with the previous report that short-term solar exposure activated the gene expression in protein processing in endoplasmic reticulum in sunlight-withered tea leaves in comparison with those of indoor-withered leaves [8]. 

### 3.4. The Metabolism Cascades of Flavor Constitutes in Different-Treated Tea Leaves

Withering is the initial step of oolong tea production, during which biochemical transformations occur to some flavor compounds such as amino acids, flavonoids, and volatile compounds [39,40]. Two different withering approaches, solar and indoor withering, were applied to the oolong tea leaves in this study. Different from indoor withering, the extra UV irradiation combined with strong sunlight in solar withering aggravates the intracellular redox imbalance caused by dehydration in the postharvest tea leaves, then the superfluous ROS would be depleted by reduced glutathione (GSH) as part of the antioxidant barrier that restrains excessive oxidation in sensitive cellular components [41], contributing to the slight decrease in GSH and in solar-withered leaves compared to indoor-withered leaves, as shown in Figure 5. GSH is biosynthesized from glutamic acid (Glu) and cysteine (Cys) via the intermediate γ-glutamylcysteine (γ-Glu-Cys) catalyzed by glutamate-cysteine ligase (gshA) as well as glutathione synthase (GSS) in plants [42]. Apart from GSH synthesis, Glu is a biosynthesis hub connecting to downstream flavoring compounds including 5′-nucleotides and alkaloids through purine/pyrimidine metabolism (Figure 5). The 5′-nucleotides acting as flavor enhancers, synergistically increase the umami taste of glutamic acids [43,44]. The bitter and astringent taste of tea brewing is associated with the accumulation of xanthine alkaloids, mainly caffeine, theobromine, and theophylline [45]. In this study, the contents of 5′-nucelotides were more abundant in SW than in the IW sample (Figure 5), suggesting that the solar withering potentially consolidates the umami taste by retaining the 5′-nucelotides in tea leaves; further, SW sample had the least accumulation of xanthine alkaloids, indicating the bitterness or astringency would be attenuated after sunlight withering. 

As a non-proteinaceous amino acid, L-theanine (Thea) is present in the greatest amount (>60%) of the free amino acids in tea plants. Thea is predominantly synthesized from Glu and ethylamine in roots of tea plants [46], and will be rapidly catabolized under the strong light exposure [35]. In this study, Thea was higher in the IW sample than in SW samples (Figure 5), likely ascribing to the solar irradiation accelerating the hydrolysis of Thea [47]. Conversely, γ-aminobutyric acid (GABA), another non-proteinaceous amino acid, was highly accumulated in SW than in the IW sample (Figure 5), as GABA in plants plays an important role in antioxidant scavenging, redox regulation, osmoregulation, and stress resistance when facing adverse conditions [48]. Additionally, the abundance of valine (Val), leucine (Leu), lysine (Lys), arginine (Arg), proline (Pro), asparagine (Asn), glutamine (Gln), phenylalanine (Phe), and tryptophan (Trp) was lower in both SW and IW samples in comparison with those in fresh leaves, implying that the dehydration occurring in the withering step reduced the accumulation of these amino acids, which is a slight difference from the observation that most of the amino acids were boosted after withering [49], while this inconsistence could be attributed to the long-time withering (>3 h) induced hydrolysis of proteins in the previous studies. 

The compounds and proteins involved in the flavonoid metabolism accounted for the dominant distinction between the withered leaves (SW, IW) and fresh leaves (FL) according to the pathway analyses in the non-volatiles (Figure 2F) and proteomic dataset (Figure 4F,G). A dozen flavonoids, abundant in fresh leaves, were significantly reduced after 30 min of withering as presented in Figure 5. This observation is congruous with the previous report that flavonoids counteracted the excessive ROS in plants to maintain redox homeostasis when exposing to diverse stresses [50]. Specifically, the catechins, including catechin, epigallocatechin, and epicatechin gallate, decreased after withering, while the contents of theaflavin derivatives rose in SW or IW samples (Figure 5), probably because catechins in the tea leaves are enzymatically oxidized to yield the complex mixtures of theaflavin derivatives over withering step [51]. 

The volatile terpenoids (VTs) and volatile phenylpropanoids/benzenoids (VPB) are two major sources of characteristic aromas of oolong tea, the moisture content of fresh tea leaves decreases and the fruity or floral aroma is formed gradually attributing to the accumulation of characteristic VTs or VPBs during withering [15,40]. Most of the VTs identified as the DRVs in this study decreased after withering, while the α/β-ionones significantly accumulated in SW sample (Figure 5). This observation is consistent with the reports that α- or β-ionone were markedly enriched in the solar-withered step and contributed to the floral fragrance of teas [52,53,54]. The contents of VPBs showed various changes in different samples in this study. The fruity and floral volatiles methyl benzoate, benzyl alcohol, benzyl acetate, phenylethanol, and phenylacetaldehyde accumulated in solar-withered tea leaves, while the abundance of most of VPBs in IW sample showed no significant accumulation after indoor withering (Figure 5), suggesting that the solar withering would aid in forming the characteristic fragrance of oolong tea. The similar result was also reported in the study on the leaves exposed to UV, in which the short-time (≤2 h) UV irradiation induced the accumulation of VPBs when comparing to the leaves in dark [55]. 

### 3.5. The Influence of Dynamic Changes in Metabolites on the Flavor of Withered Leaves 

The taste attributes of different tea samples, including bitter, astringency, aftertaste-astringency, and umami, are presented in Figure 6. Generally, the intensity of the bitter taste of SW and IW samples was significantly decreased compared to that of fresh leaves, which was attributed to the lower accumulation of bitter flavoring compounds such as Val, Leu, Lys, Arg, Trp, Phe, alkaloids, and flavonoids [35,39] in SW and IW than in CK. Moreover, the components that possess strong bitter or astringency tastes, such as catechins, were partly decomposed or oxidized into polymers during the withering (Figure 5), resulting in the dramatic reduction in astringency and aftertaste-astringency in SW and IW (Figure 6). Thea is the major source of the umami taste in tea, whose content would decrease along with the oolong tea production since it can barely biosynthesize in leaves [56], leading to the reduction in umami taste in SW (Figure 6). Notably, as the enhancers of the umami palate, the 5′-nucleotides were retained in solar-withered leaves, suggesting the umami intensity could be amplified through synergistic effect with Thea, Glu, and Asp, whereas further investigation is still needed for better understanding on the metabolism of 5′-nucleotides over the oolong tea production process. In this study, the bitterness or astringency tastes were lowered, and a few floral compounds were initially enriched, although the umami palate faded away slightly after solar- or indoor withering during oolong tea production, therefore the accumulation of the volatile/non-volatile metabolites over the withering step provided the solid foundation for the formation of the characteristic flavor of oolong tea over the following producing process. 

## 4. Conclusions

In this study, the volatile/non-volatile metabolites and proteins from different tea samples were profiled to reveal the flavor compounds conversion or accumulation induced by different withering approaches. The joint protein–metabolite analysis showed that the multiple stresses occurred in solar- or indoor withering induced the alterations of biochemical conversions upon the harvesting of tea leaves. The significant reductions in flavonoids, xanthine alkaloids, and several amino acids, contributed to the alleviation of bitter or astringent tastes of withered leaves, although the decomposition of Thea resulted in the loss in umami flavor. Moreover, the fruity or floral aromas, especially VTs and VPBs, were retained or accumulated in the solar withered leaves, promoting the formation of characteristic fragrance of oolong tea. The results described in this work confirm that the solar withering is more conducive to the formation of characteristic quality of oolong tea than the indoor withering. Our study provides a novel insight into the mechanisms underlying the relationship between the metabolic conversions and flavor formation during the withering step of oolong tea production. Further investigation on the possible conversions or enhancement of those flavor compounds induced by solar withering in the subsequent processing steps will be conducted to determine the ultimate contribution of solar withering to the final flavor quality of oolong teas. 

## Figures and Tables

**Figure 1 foods-11-03601-f001:**
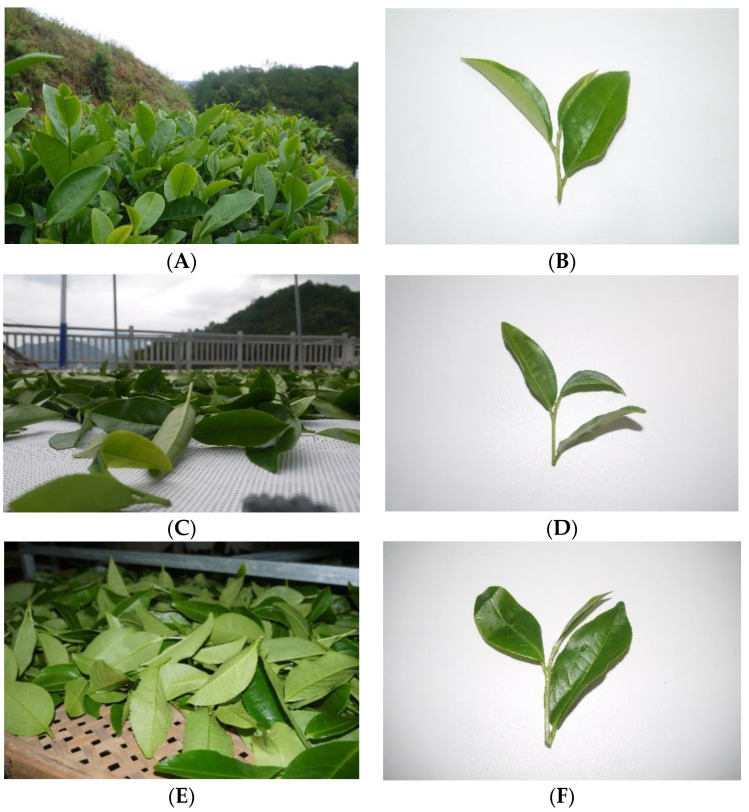
The appearance of the oolong tea samples of fresh leaves (**A**,**B**), solar-withered leaves (**C**,**D**), and indoor-withered leaves (**E**,**F**).

**Figure 2 foods-11-03601-f002:**
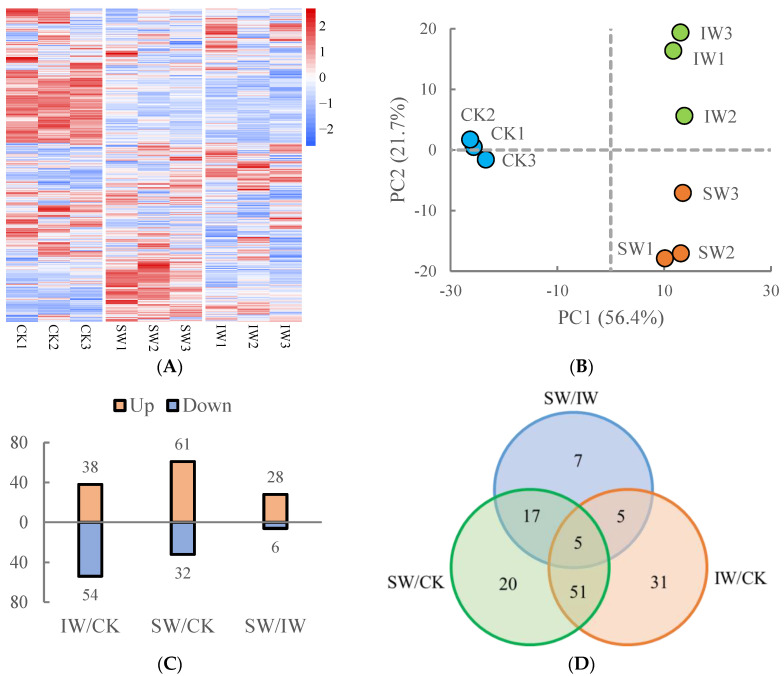
The profiles of non-volatile metabolites from different oolong tea samples. (**A**) The heatmap of the identified metabolites. The heatmap was plotted using log_2_-(fold change) values for the normalization of average abundance of the metabolomic dataset. (**B**) The principal component analysis score plot of the tea samples. (**C**) The number of differentially regulated non-volatiles (DRNs) in paired comparisons. (**D**) The Venn diagram of the DRNs from various paired comparisons. (**E**) The heatmap of shared DRNs in both SW/CK and IW/CK groups. (**F**) The pathway clustering analysis on the shared DRNs in both SW/CK and IW/CK groups. (**G**) The heatmap of DRNs in the SW/IW group. (**H**) The pathway clustering analysis on the shared DRNs in both SW/IW group.

**Figure 3 foods-11-03601-f003:**
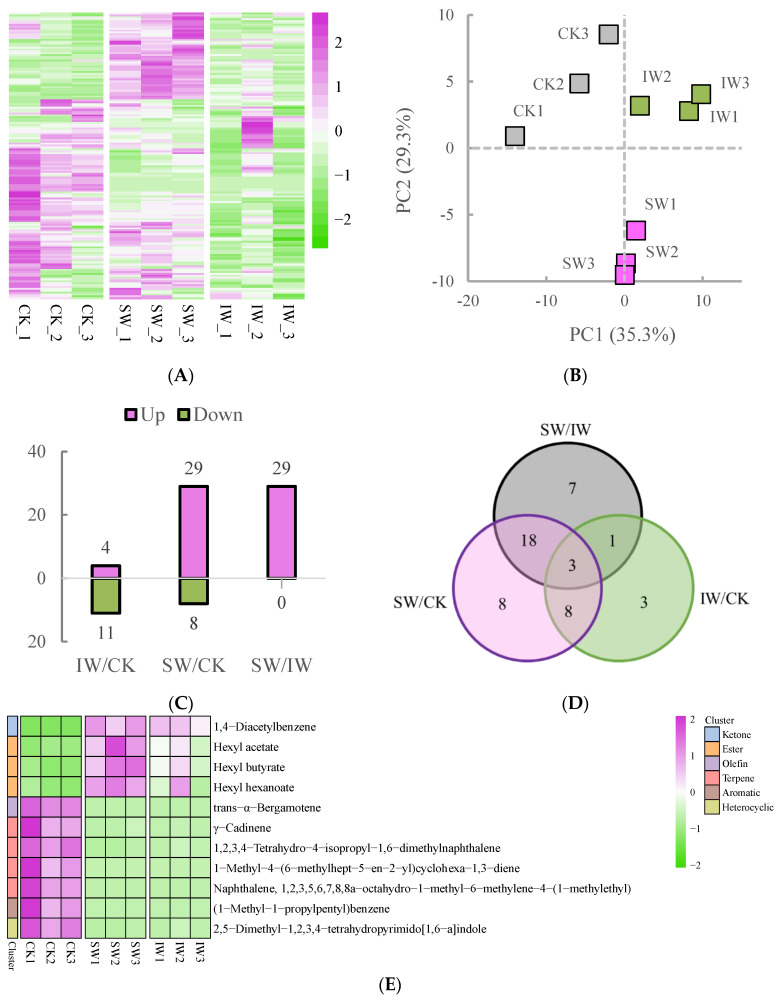
The profiles of volatile metabolites from different oolong tea samples. (**A**) The heatmap of the identified metabolites. The heatmap was plotted using log_2_-(fold change) values for the normalization of average abundance of the metabolomic dataset. (**B**) The principal component analysis score plot of the tea samples. (**C**) The number of differential regulated volatiles (DRVs) in paired comparisons. (**D**) The Venn diagram of the DRVs from various paired comparisons. (**E**) The heatmap of shared DRVs in both SW/CK and IW/CK groups. (**F**) The heatmap of DRVs in the SW/IW group.

**Figure 4 foods-11-03601-f004:**
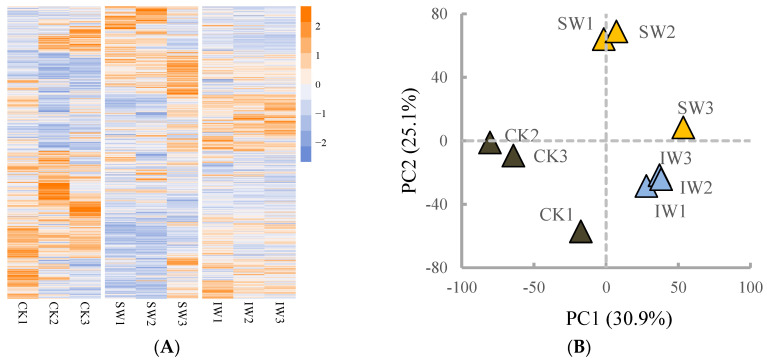
The proteomic profiles of different tea samples. (**A**) The heatmap of different tea samples based on proteomic dataset. (**B**) The principal component analysis score plot of different tea samples based on proteomic profiles. (**C**) The number of differential regulated proteins (DRPs) in paired comparisons with the criteria of fold change > 1.5 or <0.67, and false discovery rate < 0.05. (**D**) The Venn diagram of the DRPs from various paired comparisons. (**E**) The shared DRPs in both SW/CK and IW/CK groups. (**F**) The top 10 enrichment analysis pathways on the DRPs in SW/CK. (**G**) The top 10 enrichment analysis pathways on the DRPs in IW/CK. (**H**) The heatmap of DRPs in the SW/IW group. (**I**) The top 10 enrichment analysis pathways on the DRPs in SW/IW group.

**Figure 5 foods-11-03601-f005:**
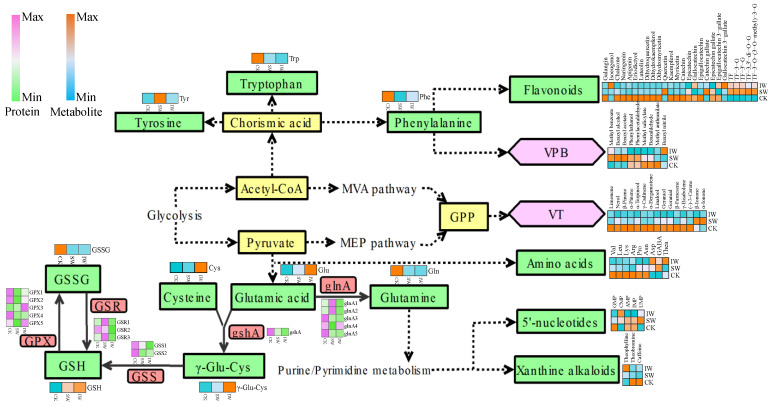
The primary dynamic changes in the metabolites in different treated oolong tea leaves. This flow chart is derived from KEGG pathway database (https://www.kegg.jp/kegg/pathway.html, accessed on 20 July 2022) and redrawn from our previous study (Reprinted/adapted with permission from Ref [18]). The heatmap plots represent the abundance of metabolites or proteins in CK, SW, and IW as labeled. The metabolites in green boxes were non-volatile compounds, and the metabolites in pink boxes were volatile compounds. MVA: mevalonic acid; MEP: methylerythritol phosphate; VPB: volatile phenylpropanoids/benzenoids; VT: volatile terpenoids; TF: theaflavin; GABA: γ-aminobutyric acid; GPP: geranyl diphosphate; γ-Glu-Cys: γ-glutamyl-cysteine; GSH: reduced glutathione; GSSG: glutathione disulfide; TF-3-G: theaflavin-3-gallate; TF-3′-G: theaflavin-3′-gallate; TF-3,3′-di-O-G: theaflavin-3,3′-di-O-gallate; TF-3-O-(3-O-methyl)-3-G: theaflavin-3-O-(3-O-methyl)gallate-3-gallate; GMP: guanosine 5′-phosphate; CMP: cytidine-5′-monophosphate; AMP: adenosine 5′-monophosphate; IMP: inosine 5′-monophosphate; UMP: uridine 5′-monophosphate; glnA: glutamine synthetase; gshA: glutamate-cysteine ligase; GSS: glutathione synthase; GSR: glutathione reductase; GPX: glutathione peroxidase.

**Figure 6 foods-11-03601-f006:**
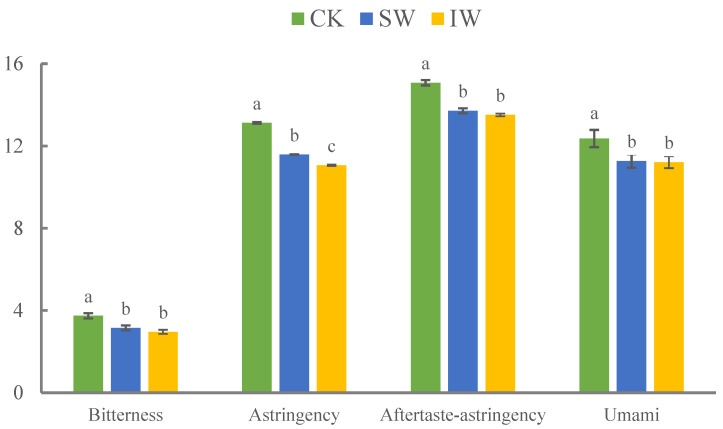
The histogram plot of E-nose analysis on the CK, SW, and IW samples. Redrawn from our previous study ((Reprinted/adapted with permission from Ref. [18]).

## Data Availability

The data are available from the corresponding author.

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
