# Peer review of "The Impact of Different Withering Approaches on the Metabolism of Flavor Compounds in Oolong Tea Leaves"

_foods, 2022, doi:10.3390/foods11223601_

Round 1

Reviewer 1 Report

1. The first sentence of the abstract should revise, it is confusing to the readers

2. In the abstract try to present which approach is better at the ending.

3. In the introduction, need to give background information about the oolong tea. Its economics, importance, and processing also need to add to the introduction

4. In the introduction what is the  work done up to know related to this work

5. what is the importance of withering in oolong tea? why selected this one only for the study should address in the introduction.

6. Why solar- and indoor-withering treatments only selected? why not any others?

7. usually the last few sentences of the introduction should be the major objective. 

8. I hope Section 2.1 should be "Tea leaf processing"

9. The websites should not give as the reference check lines 165 and 230

10. In section 2.5 add clear procedure and parameters in the determination

11. In line 176 why mentioned as biological replication? can you mention it as replication only? 

12. In the results section some places few lines are given about the procedures, which can delete. Like line number 196 and 197. 

13. In line 233, provide some detailed information on the one & flavonol biosynthesis, glutathione metabolism, and their role in flavor development

14. In line 243 can elaborate on the effect of the most possible conditions for the stress and their relation for metabolite productions clearly. 

15. Line 298, write detailed information on the role of study conditions and the development of β-ionone in the Oolong tea. 

16. In line number 312 and 313 discuss detailed about the UV irradiation, heat, and dehydration stress in the development of different components

17. In Figure 6, it is observed that SW and IW are lower why?

18. In each image title all the abbreviations should be defined

19. In the conclusions give what oolong withering approach is best.

20. The references numbered are mentioned double in the reference section check once. 

4.  

Author Response

Thanks for your comments concerning our manuscript, these comments are all valuable and helpful for revising and improving our work. We have studied all comments carefully and made conscientious correction. Revised portion are marked in red in the trackable mode in Word text.

Reviewer 2 Report

The study is novel and investigates the metabolomics of Oolong tea leaves. The authors mention the changes in components due to exposure to sunlight and the indoor environment.

 However, it would be possible for the authors to highlight the importance of the metabolomic study in the treatment proposal for this type of products.

Author Response

(The authors gave the same response as above.)

Reviewer 3 Report

As far as I am concerned the manuscript is well written. The subject area of research is significant in knowledge development. The introduction is interesting and correct. The aim of the research is clearly presented. The results of the research were presented very well.  The presented list of results is very interesting and needed another. The discussion of the results is good. The graphical presentation of the results is very interesting and clear.

Only the caption under Figure 5 has too large a space.

Author Response

(The authors gave the same response as above.)
